# An indirect method to monitor the fraction of people ever infected with COVID-19: An application to the United States

Miguel Sánchez-Romero[1], Vanessa di Lego[1]*, Alexia Prskawetz[1,2], Bernardo L. Queiroz[3]

**1** Wittgenstein Centre for Demography and Global Human Capital (IIASA, OeAW, University of Vienna), Vienna Institute of Demography/Austrian Academy of Sciences, Vienna, Austria, **2** Institute of Statistics and Mathematical Methods in Economics, TU Wien, Vienna, Austria, **3** Universidade Federal de Minas Gerais, Cedeplar, Belo Horizonte, State of Minas Gerais, Brazil

* Vanessa.DiLego@oeaw.ac.at

**Data Availability Statement:** The data and codes necessary to replicate the study have been uploaded to the following OSF repository address:

## Abstract

The number of COVID-19 infections is key for accurately monitoring the pandemics. However, due to differential testing policies, asymptomatic individuals and limited large-scale testing availability, it is challenging to detect all cases. Seroprevalence studies aim to address this gap by retrospectively assessing the number of infections, but they can be expensive and time-intensive, limiting their use to specific population subgroups. In this paper, we propose a complementary approach that combines estimated (1) infection fatality rates (IFR) using a Bayesian melding SEIR model with (2) reported case-fatality rates (CFR) in order to indirectly estimate the fraction of people ever infected (from the total population) and detected (from the ever infected). We apply the technique to the U.S. due to their remarkable regional diversity and because they count with almost a quarter of all global confirmed cases and deaths. We obtain that the IFR varies from 1.25% (0.39–2.16%, 90% CI) in Florida, the most aged population, to 0.69% in Utah (0.21–1.30%, 90% CI), the youngest population. By September 8, 2020, we estimate that at least five states have already a fraction of people ever infected between 10% and 20% (New Jersey, New York, Massachussets, Connecticut, and District of Columbia). The state with the highest estimated fraction of people ever infected is New Jersey with 17.3% (10.0, 55.8, 90% CI). Moreover, our results indicate that with a probability of 90 percent the fraction of detected people among the ever infected since the beginning of the epidemic has been less than 50% in 15 out of the 20 states analyzed in this paper. Our approach can be a valuable tool that complements seroprevalence studies and indicates how efficient have testing policies been since the beginning of the outbreak.

## Introduction

As of September 14[th], the number of confirmed cases and deaths in the U.S. corresponded to almost a quarter of all global cases and deaths, with over 6.5 million cases and 194,079 deaths,

https://osf.io/c26p7/. The DOI number is 10.17605/OSF.IO/C26P7.

**Funding:** The author(s) received no specific funding for this work.

**Competing interests:** The authors have declared that no competing interests exist.

respectively. According to the COVID-19 Data Repository by the Center for Systems Science and Engineering (CSSE) at Johns Hopkins University, the regional diversity in the U.S. is also overwhelming, ranging from 699,909 cases in California to 1,624 in Vermont and 32,957 deaths in New York to 37 in Wyoming, with a variance that reaches almost 30% between the country average and New York, in terms of the share of national fatalities [1–3]. The within-country differences are linked not only to the complex interplay of state-specific demographic and socioeconomic characteristics [4–7], but also to the inconsistent testing by region over time [8], the differential adoption of non-pharmaceutical interventions, and the timing with which each region was hit, which result in a large uncertainty on the true number of people infected [2, 9, 10]. This has led researchers to often rely on epidemiological models in order to estimate the infection fatality rate (IFR) from the inferred total number of infections in different scenarios [11–17].

More recently, the increase in availability of seroprevalence studies has enabled to use information on antibody prevalence to retrospectively assess previous infections, complementing existing models and improving the estimation certainty on the number of infections [18–22]. There are many current ongoing waves of large seroprevalence studies in the U.S., including large geographic studies, community-level studies, and studies in special populations, with each category providing complementary and important information on the extent of past infections and levels of antibodies in the population [23]. However, despite being considered as the best-practice resource to track how the prevalence of infections has evolved through time, specialists stress that it is yet not advisable to use seropositivity tests as a standalone tool to make decisions about future susceptibility to SARS-CoV-2 exposure, since evidence is still insufficient to correlate a positive serological test to immunity against the virus [24–27]. Lastly, representative seroprevalence studies require random sampling, which at a national level is resource-intensive, especially in a large country as geographically diverse as the U.S. [28].

All of the aforementioned aspects make it challenging to accurately estimate the fraction of people ever infected, which remain a crucial information to monitor the evolution of the pandemics and its aftermath. As governments try to manage the massive socioeconomic consequences of the lockdown [29–31] and while effective pharmaceutical interventions are not yet available, resources become even more limited, demanding the widest possible mix of alternatives to monitor the evolution and consequences of the pandemic in a large-scale. This paper contributes to this analysis by proposing a method to indirectly estimate the fraction of people ever infected (from the total population) and the fraction of people detected (from the total population ever infected). The approach combines IFRs that are estimated through a SEIR (susceptible-exposed-infected-removed) model with deaths and reported case-fatality rates (CFR) reported by countries. We first estimate the IFRs by fitting a SEIR model that takes into account demographic characteristics, such as the age distribution of the population and underlying age-specific COVID-19 and non-COVID-19 mortality rates [32–34], through a Bayesian melding approach [35]. The main advantage of using the Bayesian melding is to better manage the high degree of uncertainty in COVID-19 data, since it derives the distribution of the set of parameters that best replicates the observed evolution of deaths by using information from the model and the data. Secondly, we develop an indirect estimation technique to estimate the fraction of people ever infected (from the total population) and detected (from the ever infected). We show that because both IFR and CFR, by definition, depend only on the probability of dying conditional on being infected, it is possible to relate the properties of those two measures in order to indirectly estimate the fraction of people ever infected (from the total population) and the fraction of people detected (from the total population ever infected) only accounting for deaths. Because of the characteristics and timing of both the pandemic evolution and the reported CFRs by countries, it is important to satisfy two criteria in order to

ensure that the estimates are meaningful, accurate, and reflective of the situation being portrayed. First, a stable average rate of growth in the death toll must have been reached, so that it is guaranteed that most infected people are either recovered or dead. Second, a minimum amount of deaths must have occurred to enable that COVID-19 deaths are distributed across most age groups (for further details on these conditions see the Material and methods section below and S3 and S4 Sections in S1 File). As long as a large fraction of infected people are either dead or recovered and that the number of deaths has been large enough to guarantee that COVID-19 deaths are distributed across most age groups, our model can be applied to any country/region for which demographic information on age-specific (non-COVID-19) mortality rates, population age structure, and reported COVID-19 CFRs are available. After meeting both criteria, we could apply our model to estimate the fraction of people ever infected across 20 U.S. states. One main advantage of the proposed method is to report uncertainty levels to the number of infected people. As has been previously indicated in other studies, the age distribution of the population is important to explain the variation in the estimated IFRs and the reported CFRs across populations [32–34, 36]. For Florida, the most aged population in our study, we estimate an IFR of 1.25% (0.39–2.16%, 90% CI), which is almost twofold higher than that of the youngest population of Utah, with an estimated IFR of 0.69% (0.21–1.19%, 90% CI). According to the CFRs reported by authorities as of September 8, 2020, our model estimates that at least five states have a fraction of people ever infected over 10% (New Jersey, New York, Massachussets, Connecticut, and District of Columbia). The state with the highest estimated fraction of people ever infected is New Jersey with 17.3% (10.0, 55.8, 90% CI). Moreover, our results indicate that with a probability of 90 percent the fraction of detected people among the ever infected has been less than 50% in 15 out of the 20 states analyzed in this paper since the beginning of the epidemic. This implies that despite the increase in testing capacity, most states are still struggling to efficiently detect infected persons, suggesting that more intensive or targeted efforts are needed for controlling the pandemics.

## Material and methods

As a first step, we estimate the IFRs for the U.S. states using an epidemiological SEIR model that (a) accounts for the age distribution of the population, (b) explicitly models the differential effect of non-COVID-19 mortality by age, and considers (c) the differential effect of COVID-19 mortality by age. The model focuses on the number of deaths from COVID-19 since this information is more reliable than the number of infections, despite the fact that it can also be subject to under-reporting [37] as well as to over-reporting, due to competing causes and indirect effects on death [38–40]. However, since the data on deaths is likely incomplete, we implement the Bayesian melding method [35], which provides an inferential framework that takes into account both model's inputs and outputs. Bayesian melding has already been applied in epidemiological settings such as to study the HIV/AIDS prevalence [41–44] by UNAIDS. The model proposed here is different from the model used by CDC to monitor the pandemic. CDC uses a mathematical model based on different scenarios to evaluate different mitigation strategies. To control in our estimates for the unknown number of underreported infected cases, the SEIR model includes an adjustment factor for accurately deriving the IFR. Age-specific mortality by state is from the United States Mortality Database [45] and single-year age population for each U.S. state is retrieved from the July 2019 version from the U.S. Census Bureau (We downloaded the table Single Years of Age and Sex Population Estimates: April 1, 2010 to July 1, 2019—CIVILIAN (SC-EST2019-AGESEX-CIV)). Our estimated IFR is a linear combination of the IFR across 3000 SEIR models for each U.S. state that best replicates the

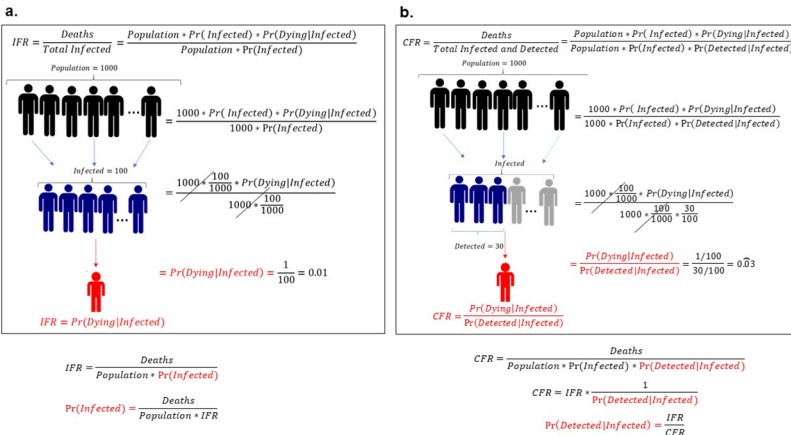

**Fig 1. Definition and properties of CFR and IFR.** The CFR and IFR both depend on the probability of dying once a person is infected, but not on the number of persons infected. Panel **a** and **b** show a hypothetical situation where in a population of 1000 persons, 100 get infected, with 1 death among all the infected. The difference is that on panel **b** only 30 out of the 100 persons who are infected are detected. This changes the denominator, making the CFR larger than the IFR. Below the panel we show how the probability of ever being infected (or the fraction of infected persons) and the probability of being detected among the population ever infected (or the fraction of detected persons), respectively, can be defined using the estimated IFR and the CFR.

data. For more details on the specification of the Bayesian melding model to derive the IFR consult S1 and S2 Sections in S1 File.

As a second step, after estimating the IFR by state, we show how we can indirectly estimate both the fraction of people ever infected from COVID-19 across U.S. states and the fraction among the infected that are detected. The key for performing this indirect estimation relates to the definition of the IFR and the CFR. We use the definition of the IFR and CFR as supported by the World Health Organization (WHO) and other specialists [11, 46]. WHO provides a definition of the CFR when the pandemic is not controlled, which is defined as the "ratio between the number of deaths from disease to the sum of the number of deaths from disease and the number of recovered from disease" [47]. When the disease is controlled, this definition coincides with an alternative definition of CFR, namely the "ratio between the number of deaths from disease to the number of confirmed cases of disease". Since the results calculated in this manuscript are only done once that the disease is controlled, we apply the latter definition. Fig 1 shows an illustration of how we can combine the properties of both measures. For the sake of expositional clarity, in the following explanatory example we abstract from age differences, but in the results all yielded estimates include age-specific differences. The definition of IFR–number of total deaths divided by the number of total infections, stated in the top left of panel Fig 1a, can be written as the product of the population size, the probability of being infected and the probability of dying given that one is infected–the numerator of the IFR–divided by the product of population size and the probability that one is infected–the denominator of the IFR. In a hypothetical situation presented in panels Fig 1a and 1b, the number of people infected—which is the product of the total population size, (here 1000 persons assuming that everyone is susceptible) and the probability of being infected (100/10000),—does not influence the probability of dying (1/100) once a person is infected, since these values are present both in the numerator and the denominator, offsetting each other. Hence, it is possible to define the IFR as the probability of dying conditional on being infected, which is the first property that has important implications for how we understand the COVID-19 dynamics of infections and deaths, as we will discuss in more detail below. In the bottom part of the panel, we rearrange

the equations in panel Fig 1a in order to define the probability of being infected (fraction of persons infected) as the ratio between deaths and the product of population and the estimated IFR from our model. Since we have shown that, by definition, the IFR is not affected by the probability of being infected, but only by the probability of dying once you get infected, we can use the estimated IFR from our model without accounting for the state-specific evolution of infections, as it does not add any further explanatory power to the model. Similarly, on panel Fig 1b, we start with the definition of the CFR as the ratio of the total number of deaths to the total infected and detected. This means that the only difference between the CFR and the IFR is that in the denominator we have to multiply the total infected by the probability of being detected, which is the number of persons a country can detect among all the infected cases. We apply the same hypothetical situation as in panel Fig 1a, but now we assume for illustrative purposes that only 30 out of the 100 infected persons are detected. We not only reach the same conclusion that the probability of being infected and the total population size do not affect the probability of dying, but we can define the CFR in terms of the IFR divided by the probability of being detected conditional on being infected, since the only difference between these two indicators is the denominator of the former being affected by the probability of detecting persons among the infected. Rearranging the terms in the bottom right-hand side of panel Fig 1b, we can define the probability of being detected conditional on being infected by the ratio of the IFR and the CFR.

The relationship between these two quantities and the fact that they are independent of the number of infected cases allows us to indirectly estimate the fraction of people ever infected and the fraction of people detected among the ever infected by combining the estimated IFR from our model and the CFR reported from government officials. Therefore, it is not necessary that our model accounts for the evolution of the total number of infected people at each point in time, which validates our simulation strategy.

Before we apply this technique to U.S. states, we perform an external validation to test the sensitivity of the approach. We show that our indirect estimation method is capable of replicating existing studies of seroprevalence and fraction of people ever infected, which are the best-practice attempts at approximating the fraction of people ever infected across different regions and countries. We focus our analysis on first wave seroprevalence studies that include all age groups of the population, are randomized, and representative of the population. This is important since many seroprevalence surveys focus on specific segments of the population and exclude some age groups, which does not allow to draw inferences at the population level. The first seroprevalence studies that were carried out and met those criteria are the ones from Brazil [22], Spain [48] and those conducted for Connecticut, Missouri, New York City (NYC), and the state of New York (data from the National Center for Immunization and Respiratory Diseases (NCIRD), Division of Viral Diseases) [49]. S6 Fig **and S5 Table** in S1 File show that our indirect estimation replicates well the fraction of people ever infected among the total population reported in these seroprevalence studies. After showing that our approach is capable of replicating existing seroprevalence studies, we apply our model to estimate our two measures across U.S. states: (i) the estimated fraction of people ever infected and (ii) the fraction detected among the infected. In order to account for the fact that disruptions in how the CFR is reported may lead to artificial fluctuation in the CFR [9], the states analyzed must satisfy two criteria. First, the state must have a stable death toll. We assume that a stable death toll is reached when its average rate of growth during the last month is less than 0.5%. This condition guarantees that most infected people are either recovered or dead. Second, we exclude states with less than 500 reported deaths from COVID-19 in order to guarantee that COVID-19 deaths are distributed across most age groups. This last criterion excludes 15 states (Alaska, Hawaii, Idaho, Kansas, Maine, Montana, Nebraska, New Hampshire, North Dakota, Oregon,

South Dakota, Utah, Vermont, West Virginia, Wyoming) in which less than 500 COVID-19 deaths have been reported. As a result, by combining epidemiological data taken from the COVID-19 Data Repository by the Center for Systems Science and Engineering (CSSE) at Johns Hopkins University on September 8, 2020 and the distribution of IFR estimated with our model, we can estimate the fraction of people ever infected among the total population and the fraction detected among the ever infected for 20 states.

## Results

### Estimated IFRs by state

The first set of results (based on the SEIR model) are the estimated IFRs (the ratio between the total deaths to total infected people) for 50 states and D.C. in the U.S., as presented in Fig 2.

Part of the East Coast and D.C area are zoomed in for better visualization. The state with the highest estimated IFR is Florida with 1.25% (0.39–2.16%, 90% CI) followed by Maine 1.23% (0.38–2.13%, 90% CI), Hawaii 1.21% (0.37–2.08%, 90% CI), West Virginia 1.17% (0.36–2.02%, 90% CI), and Vermont 1.16% (0.36–2.01%, 90% CI). The state with the lowest estimated IFR is Utah with 0.69% (0.21–1.19%, 90% CI), followed in reverse order by Alaska 0.75% (0.23–1.30%, 90% CI), Texas 0.79% (0.25–1.37%, 90% CI), District of Columbia 0.80% (0.25–1.39%, 90% CI), and Georgia 0.86% (0.27–1.48%, 90% CI). The terms in parenthesis show the 90 percent credible intervals, which are the intervals within which the IFR for each state falls with a probability of 90 percent. Thus, with a probability of 90 percent, the estimated IFR ranges from 0.36 to 2.16 percent in Florida, the state with the oldest age structure, and from 0.21 to 1.19 percent in Utah, the state with youngest age structure. Nonetheless, given the high uncertainty around the estimated IFR, we also provide in **S2 Table** in S1 File the 68% credible intervals.

### Indirect estimation of fraction of people ever infected and fraction detected among the ever infected

Combining the CFRs reported by countries from the COVID-19 Data Repository by the Center for Systems Science and Engineering (CSSE) at Johns Hopkins University on September 8,

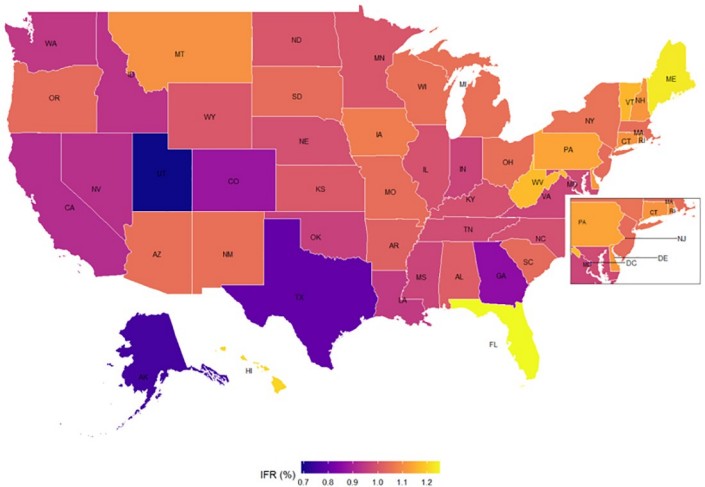

**Fig 2. Estimated Infection Fatality Rate (IFR) by state.** This refers to the infection fatality rate (the ratio between the total deaths to total infected people) See more details in **S1 Section** in S1 File for the process in estimating the IFR.

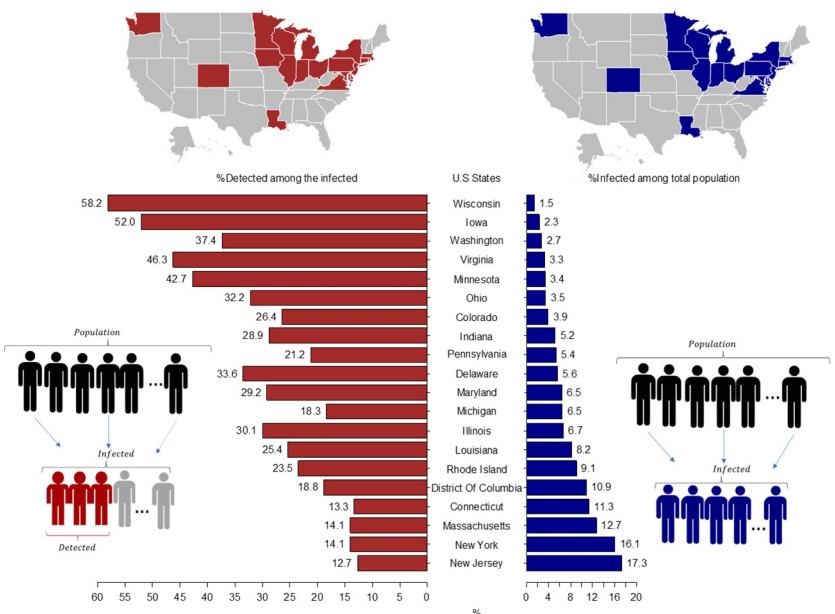

**Fig 3. Estimated fraction of people ever infected among the total population and fraction of people detected among the ever infected across states with stable CFR as of September 8, 2020.** The left part of the graph concentrates on the % of detected among all the infected individuals, which are in the bottom-left diagram depicted as the highlighted (red) individuals divided by all the infected. The right-hand side shows the % of infected among all the population, so depicted in the diagram in the bottom-right as all the individuals in the second part of the diagram (blue) divided by the first part (total population). On the top we highlight the regional distribution of those indicators in the U.S. map. Credible intervals for each bar are provided in S3 and S4 Tables in S1 File.

2020, with the estimated IFR from our model, we report in Fig 3 the average values of the fraction of people ever infected among the total population and the fraction detected among the ever infected for 20 states. We estimate that at least in five states the fraction of people ever infected is above 10%. As the right-hand side of Fig 3 shows, the state with the highest estimated fraction of people ever infected is New Jersey with 17.3% (10.0–55.8, 90% CI), followed by New York 16.1% (9.3–51.9, 90%CI), Massachusetts 12.7% (7.3–40.9, 90% CI), Connecticut 11.3% (6.6–36.5, 90% CI), and the District of Columbia 10.9% (6.3–35.0, 90% CI). See the S3 Table **in** S1 File for all values with respective credible intervals and the dates at which this fraction is estimated for.

In addition, we estimate the fraction of people ever infected who are detected or the probability of being detected among the ever infected. This calculation is useful for assessing the effectiveness of testing policies and also for detecting regions or groups that need additional testing. To do so, we use the definition of the IFR and the CFR as mentioned in Fig 1. The left-hand side of Fig 3 and **S4 Table in** S1 File show that the state with the highest estimated fraction of individuals detected among the ever infected is Wisconsin 58.2% (18.1–100.0, 90% CI), followed by Iowa with 52.0% (16.9–90.0, 90% CI), Virginia 46.3% (14.4–80.1, 90% CI), Minnesota 42.7% (13.3–73.9, 90% CI), and Washington 37.4% (11.6–64.7, 90% CI). Nonetheless, the data used for Wisconsin and Iowa is before September 8, 2020, which suggests that the current fraction of individuals detected among the ever infected is lower. For the remaining 15 states, we estimate that with a probability of 90 percent the fraction of detected people among the ever infected has been less than 50%. For these states, this result implies that unless other non-pharmaceutical interventions such as social distancing, the use of masks, and hygiene measures are implemented, it is expected that the spread of the SARS-CoV-2 virus will continue

(24). This highlights the importance of alternative approaches that allow for tracking the pandemic evolution and can be applied with currently available data.

## Discussion

By recurring to the definition of the IFR and CFR, key indicators to monitor the evolution of the pandemics, we could shed light to an important feature that is often overlooked: both these indicators are conditional probabilities. As a result, the probability of dying from COVID-19 *conditional* on being infected is independent from the probability of being infected, as the conditional term states. Hence, the processes that affect the *probability* of dying *given* that a person has been infected and the *probability* of being infected in the first place are two processes that are differently affected by measures and policies such as non-pharmaceutical (NPIs) and pharmaceutical interventions (PIs). The probability of dying conditional on being infected is only affected by pharmaceutical interventions (PIs), such as vaccines or clinically proven treatment, and despite ongoing efforts to develop such interventions they are still unavailable or very limited [50–52]. On the other hand, the probability of being infected is affected by non-pharmaceutical interventions (NPIs), whose main role is avoiding the transmission or preventing people from getting infected. An equally important role of NPIs is to alleviate the burden of the healthcare system by spreading out the cases in time and allowing everyone to be treated [16, 53–55]. However, NPIs do not affect the *probability* of dying conditional on a person being infected. This is because the probability of dying once a person is infected is related to individual risk factors, which increase considerably with age and mainly underlying health issues, in particular cardiovascular conditions, diabetes and hypertension [56–58].

As a first consequence, by acknowledging the fact that both the IFR and CFR are conditional measures, and as such do not depend on non-pharmaceutical interventions, we can develop a more parsimonious epidemiological model that does not need to account for complex mitigating scenarios in order to estimate the IFR. Adding different lockdown scenarios or measures adopted by states, countries or regions adds no explanatory power to the model as the indicators we are interested in are conditional probabilities of dying that are not affected by such NPI interventions. Therefore, it is not necessary to account in our model for the evolution of the total number of infected people at each point in time, which validates our simulation strategy. A lot of remarkable work has been done on virus transmission and their effect on societies, accounting for all the complexity in mitigating scenarios [12, 17, 59, 60]. We acknowledge those models and they are of paramount importance to understanding the effects of the pandemics as they address the factors that are linked to the probability of being infected. Our aim is to contribute by using the characteristic of conditional probability of both CFR and IFR to indirectly estimate the fraction of people ever infected out of the total population and the fraction of detected among the ever infected.

Noteworthy of mention, the characteristic of conditional probabilities that we have addressed is valid as long as there is no effective pharmaceutical intervention or treatment, since this will affect the probability of dying conditional on being infected. Our model also assumes that the fatality rate for COVID-19 does not change overtime. If this assumption is not satisfied, our model would be underestimating the total number of people ever infected, which implies that our estimates as they are would be providing the lower bound of the total number of people ever infected. Furthermore, should a better and new treatment reduce the IFR, then our estimated fraction of people ever infected would also increase by the same percent. In addition, our indirect estimation of the fraction of people ever infected depends on the accuracy of the total number of deaths from COVID-19. If there is a state or country

that has systematically excluded (or included) a specific fraction of the true total deaths from COVID-19 (or included non-COVID-19 deaths), the true fraction of people infected will be this same specific fraction times higher (or lower) than the fraction of people ever infected reported in S3 Table in S1 File. We discuss in S1 File how this information can be used to adjust for these errors. Additionally, we do not consider detailed information on comorbidities and health conditions that are known to affect the probability of dying conditional on being infected [56, 58, 61]. We would need detailed information such as age-specific rates for those conditions by region, which is not only very limited but lacks comparability. Other factors such as socioeconomic and ethnic characteristics, still lack conclusive evidence on whether these features indeed increase the risk of dying once a person is infected. Research that focused on England showed that blacks relative to whites and higher levels of deprivation were associated to higher mortality among the infected [62]. Others show that these characteristics are actually linked to increased risk of COVID-19 infections, but not necessarily to higher risk of dying, mostly because these individuals are exposed through lower skilled or higher risk jobs, but present a younger age pattern in infections [63]. In the U.S. case, there is already a persisting inequality in mortality by race, so it is difficult to disentangle to what extent COVID-19 deaths will affect this pattern and if the probability of dying once infected will vary by race. As recent research showed, in order for white mortality in 2020 to reach levels already experienced by blacks, COVID-19 mortality levels would need to increase by a factor of nearly 6 [64]. In this way, it is yet inconclusive whether race affects the probability of dying once infected, despite the fact that there is more evidence that they experience a higher probability of being infected. Gender has also been pointed out as an important individual risk factor, with men presenting higher mortality risk relative to women [61, 65–67], despite women being more exposed to the risk of infection in certain contexts [68]. In this work, we perform all estimates for the total population, due to the already high level of uncertainty in current data and the unavailability of accurately reported CFRs by countries that are broken by age and sex, despite recent improvements [69, 70].

However, despite the aforementioned, it has been shown that the age distribution of COVID-19 mortality is related to all-cause mortality across different countries, suggesting that the age pattern of COVID-19 mortality reflects pre-existing health inequalities in mortality within populations [34]. Age, in itself, is associated to an increasing prevalence of comorbidities and other health conditions which increase the risk of death, which is consistent with the fact that COVID-19 mortality is higher for older individuals [71, 72]. In terms of CFR, it has been shown that the age-structure of detected cases often explains more than two-thirds of cross-country variation [36]. Hence, despite not explicitly accounting for underlying health conditions and comorbidities, as well as socioeconomic, gender and ethnic factors, because we account for age and pre-existing age-specific mortality in each U.S state, we are able to partially capture health conditions and underlying factors associated to differential mortality risks.

The advantage of our approach is that after estimating IFR it offers an indirect and complementary way to approximate the fraction of infected and detected individuals, which can be valuable in contexts where population-level seroprevalence studies are hindered by financial or time constraints. It additionally offers a benchmark with which seroprevalence tests can be compared to, in order to aid in drawing a fuller picture in the very uncertain scenario of the COVID-19 pandemic. Lastly and very importantly, it may offer and indication of whether countries have tested the population in an efficient manner. It is widely acknowledged that the more tests performed, the higher is the likelihood of detecting individuals and taking appropriate measures [73, 74]. However, since universal and frequent testing is not possible, countries

only test parts of the population. This strategy can be efficient if the testing is done properly, with randomized design; testing a lot does not necessarily imply a higher detection among the infected. Higher levels of testing may give the impression of more pandemic control, but the transmission is controlled by detecting individuals among the infected, while what is reported is the confirmed cases among all the individuals who are tested.

As currently one of the countries in the world with the highest number of cases and deaths from COVID-19, the U.S. also has been impacted in diverse ways at the geographical level. This diversity makes it even more difficult to assess the already challenging spread and consequences of the virus, demanding that regional level characteristics are adequately accounted for when designing coordinated policy responses that are effective. We have shown that because most states are detecting extremely low levels among the infected, unless that other non-pharmaceutical interventions such as social distancing, the use of masks, and hygiene measures are implemented or maintained in force, it is expected that the spread of the virus will continue, since in many of the 20 states analyzed not even half of the population has been detected.

## Supporting information

**S1 File.**
(PDF)

## Acknowledgments

We thank the Brown Bag seminar participants at UC Berkeley, IIASA-RANEPA Online Symposium: Demographic Consequences of COVID-19, ORCOS Workshop on the Optimal Control of Pandemics, IST Austria, and the CFE-CMStatistics Conference, for the helpful comments and suggestions. We particular address Adrian Raftery, Ronald Lee, Josh Goldstein, Tim Miller, Sergei Scherbov, and Ester Gonzalez-Prieto for very insightful discussions that enabled us to improve our work. We also thank Flávio Freire, Marcos Gonzaga and José Henrique Costa Monteiro da Silva for acknowledging us with city level data for Brazil. We also thank referees on an earlier draft of the paper for helping us to improve our work.

## Author Contributions

**Conceptualization:** Miguel Sánchez-Romero, Vanessa di Lego, Alexia Prskawetz, Bernardo L. Queiroz.

**Formal analysis:** Miguel Sánchez-Romero, Vanessa di Lego, Alexia Prskawetz.

**Investigation:** Miguel Sánchez-Romero, Vanessa di Lego, Alexia Prskawetz, Bernardo L. Queiroz.

**Methodology:** Miguel Sánchez-Romero, Vanessa di Lego.

**Software:** Miguel Sánchez-Romero.

**Validation:** Miguel Sánchez-Romero.

**Visualization:** Miguel Sánchez-Romero, Vanessa di Lego.

**Writing – original draft:** Miguel Sánchez-Romero, Vanessa di Lego.

**Writing – review & editing:** Miguel Sánchez-Romero, Vanessa di Lego, Alexia Prskawetz, Bernardo L. Queiroz.

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
