## [Decision Letter · Decision Letter 0]

10 Nov 2020

PONE-D-20-30049

An indirect method to monitor the fraction of people ever infected with COVID-19: an application to the United States

PLOS ONE

Dear Dr. di Lego,

Thank you for submitting your manuscript to PLOS ONE. After careful consideration, we feel that it has merit but does not fully meet PLOS ONE’s publication criteria as it currently stands. Therefore, we invite you to submit a revised version of the manuscript that addresses the points raised during the review process.

Please fully address the comments by the two reviewers and resubmit your manuscript for consideration for publication.

We look forward to receiving your revised manuscript.

Kind regards,

Sanjai Kumar

Academic Editor

PLOS ONE

Journal Requirements:

Reviewers' comments:

Reviewer's Responses to Questions

**Comments to the Author**

1. Is the manuscript technically sound, and do the data support the conclusions?

Reviewer #1: Yes

Reviewer #2: Yes

2. Has the statistical analysis been performed appropriately and rigorously? 

Reviewer #1: Yes

Reviewer #2: Yes

3. Have the authors made all data underlying the findings in their manuscript fully available?

Reviewer #1: Yes

Reviewer #2: Yes

4. Is the manuscript presented in an intelligible fashion and written in standard English?

Reviewer #1: Yes

Reviewer #2: Yes

5. Review Comments to the Author

Reviewer #1: Sanchez-Romero et al. used statistic and epidemiologic tools to indirectly estimate the fraction of people ever infected by SARS-CoV-2 (from the total population) and the fraction of people detected (from the total population ever infected) in the United States. Overall, the manuscript is well-written and the study produced valuable information about COVID-19 in the U.S.

Major comment:

Please clarify the difference of your model presented in this manuscript from the one suggested by WHO (https://www.who.int/news-room/commentaries/detail/estimating-mortality-from-covid-19) and the one used by CDC (https://www.cdc.gov/coronavirus/2019-ncov/hcp/planning-scenarios.html)

Minor comments:

Please check acronyms throughout the text. There are some acronyms spelled out more than once and others that are not spelled out in the first time they are used.

Authors should be consistent when using the definition of IFR. In the Section 1 of Supporting Information, they state that “The infection fatality rate (hereinafter IFR) is defined as the ratio between the total number of deaths from COVID-19 and the total number of infectious individuals”. On page 9, line 178, IFR is defined as “number of total deaths by the number of total infections”. Infectious individuals and infected individuals cannot be used interchangeably.

Reviewer #2: In this paper, author proposed a Bayesian SEIR model to estimate infection fatality rate (IFR), which can then be combined with reported case-fatality rate (CFR), in order to indirectly estimate the fraction of people ever infected and detected. I have the following review comments:

1. The proposed method assumes the fatality rate for COVID-19 does not change overtime. In lines 388-391, authors stated the method is valid when there is no effective treatment, since this will affect IFR. However, the fatality rate for COVID-19 in U.S. appears to decrease overtime, which may due to improved standard of care or new treatment interventions. It’s important to discuss, in the manuscript, how the proposed method will behave if the assumption of constant IFR is not satisfied. Because this is likely the case.

2. Table S1 in the Supporting Information (SI) shows the proposed method calculated point estimates for incubation period as 2.57 days, recovery period as 4.12 days, and transmission period as 1.7 days. These estimates appear to be lower than other resource indicates. For example, WHO’s COVID-19 Situation Report 73 indicates the incubation period for COVID-19 is on average 5-6 days. A study (vanKampen, June 2020) indicates the median duration of infectious virus shedding is 8 days post onset of symptoms. Please comment on how these epidemic parameters estimated from the proposed model compare to results from other studies and whether biases exist for these estimates.

3. I have following comments to the external validation results (i.e., Figure S6 and Table S5) for estimated IFR in the SI.

a. Figure S6 only shows the absolute difference between estimate and observed IFR in the right-hand side panel. It will be informative to also provide relative difference between estimated and observed IFR (i.e., relative difference=(estimated-observed)/observed).

b. The labels in Figure S6 could be misleading, especially in the right-hand side panel, where -0.1 and 0.1 represents ±10% difference in fatality rate. Please change the label in x and y axis to percentage (e.g., -10%, +10%), which is consistent with Table S5 and rest of the document.

c. For Table S5, please add 90%CI and/or 68%CI for the estimated IFR. This will illustrate how many of the observed IFRs actually fall in the credible intervals.

d. Some results in Table S5 show relatively large difference between estimated and observed IFR (e.g. Sao Luis, Brazil; Recife, Brazil; and Sao Paulo, Brazil) even with a large number of deaths. Please comment.

6. PLOS authors have the option to publish the peer review history of their article (what does this mean?). If published, this will include your full peer review and any attached files.

Reviewer #1: No

Reviewer #2: No

---

## [Author Response · Author response to Decision Letter 0]

9 Dec 2020

We would like to thank the reviewers and editor for your time, for carefully reading our paper, and providing many thoughtful and important comments. We believe that this revised version has been significantly improved as a result. Please find both below and in our Response to Reviewers .doc file our detailed addressment to each issue that was raised. Besides detailing them in our Response letter, we have highlighted where those changes were made in the tracked changes in the main Manuscript. We have also sent a second Supplementary Information file, besides the revised version, with red color markings to signal the changes that were also related to the Supplementary Information. Should further remarks be addressed, we are ready to implement any changes required.

Reviewer #1: Sanchez-Romero et al. used statistic and epidemiologic tools to indirectly estimate the fraction of people ever infected by SARS-CoV-2 (from the total population) and the fraction of people detected (from the total population ever infected) in the United States. Overall, the manuscript is well-written and the study produced valuable information about COVID-19 in the U.S.

Major comment:

1. Please clarify the difference of your model presented in this manuscript from the one suggested by WHO (https://www.who.int/news-room/commentaries/detail/estimating-mortality-from-covid-19) and the one used by CDC (https://www.cdc.gov/coronavirus/2019-ncov/hcp/planning-scenarios.html). 

 Response: We thank the reviewer for proposing this comparison. WHO does not provide a simulation model like the one developed in this manuscript. The WHO’s webpage provides a definition of the CFR when the pandemic is not controlled, which is defined as “ratio between the number of deaths from disease to the sum of the number of deaths from disease and the number of recovered from disease”. When the disease is controlled, this definition coincides with the alternative definition of CFR, namely the “ratio between the number of deaths from disease to the number of confirmed cases of disease”. Since the results calculated in this manuscript are only done once that the disease is controlled, we apply the latter definition. 

The CDC is following several different strategies to monitor the evolution of the pandemic. First, the CDC and the Office of the Assistant Secretary for Preparedness and Response have developed a mathematical model, which is based on scenarios, to evaluate the potential effects of different mitigation strategies. In this regard, our model is a more stylized model than that used by the CDC. However, our model is more sophisticated in terms of calibration due to the use of the Bayesian melding. Our estimated IFR is calculated as a linear combination of 3000 different IFRs from models that are capable of fitting well the evolution of the number of deaths. Thus, each one of these 3000 models can be considered as a scenario, which is however constrained to fit the evolution of the evolution of the number of deaths. The CDC is also monitoring the stage of the pandemic by using seroprevalence studies across US states. The results of these studies were not available when we submitted our manuscript. So, we have included in the revised version of the supplementary material Figures S8 and S9 that compare our model results to the seroprevalence results in round 1 and 4. In Fig S8 we show the absolute and relative errors between the estimated fraction of people ever infected and the US COVID-19 seroprevalence estimate by state. In Fig S9 we show the absolute and relative errors between the fraction of people ever infected and the US COVID-19 seroprevalence estimate by state. With these two Figures we can show that our model can reproduce well the prevalence estimates reported in the Nationwide Commercial Laboratory Seroprevalence estimate by state (CDC). The average absolute error between our estimates and those of the Surveys is almost zero for both rounds and the relative error between our estimates and those of the Surveys is close to 0.29 for both rounds. 

Minor comments: 

2. Please check acronyms throughout the text. There are some acronyms spelled out more than once and others that are not spelled out in the first time they are used. 

 Response: Thank you for the careful reading. We have checked the acronyms thoroughly. The first moment they are spelled out in the main text are written in the comment box of the track changes in the main file. After the first definition, we keep only the acronym throughout the text. We hope this provides a clearer read of the text; let us know should you prefer an alternative solution.

3. Authors should be consistent when using the definition of IFR. In the Section 1 of Supporting Information, they state that “The infection fatality rate (hereinafter IFR) is defined as the ratio between the total number of deaths from COVID-19 and the total number of infectious individuals”. On page 9, line 178, IFR is defined as “number of total deaths by the number of total infections”. Infectious individuals and infected individuals cannot be used interchangeably.

 Response: Thank you for carefully reading the paper. We have corrected the inconsistent definition. In our case the IFR is defined as “number of total deaths by the number of total infections”.

Reviewer #2: In this paper, author proposed a Bayesian SEIR model to estimate infection fatality rate (IFR), which can then be combined with reported case-fatality rate (CFR), in order to indirectly estimate the fraction of people ever infected and detected. I have the following review comments:

1. The proposed method assumes the fatality rate for COVID-19 does not change overtime. In lines 388-391, authors stated the method is valid when there is no effective treatment, since this will affect IFR. However, the fatality rate for COVID-19 in U.S. appears to decrease overtime, which may due to improved standard of care or new treatment interventions. It’s important to discuss, in the manuscript, how the proposed method will behave if the assumption of constant IFR is not satisfied. Because this is likely the case. 

 Response: We thank the reviewer for this very important remark and providing us with the opportunity to further clarify this point. The reviewer correctly points out that that the IFR could be changing. Should that be the case, our model would be underestimating the total number of people ever infected and thus be providing the lower bound of the total number of people ever infected. Nevertheless, since conclusive evidence regarding the efficacy of current pharmaceutical interventions or better treatment in reducing mortality is still lacking, we believe our assumptions are still valid. Additionally, it is possible that changes in the fatality rate are mainly driven by the shifting pattern of the mean age of people infected, as we could see from the shift from an older age structure in terms of infections in the first wave to a younger age structure in infections in the summer, while starting in autumn it is again affecting older persons. In order to clarify the implications of our assumptions, we have added this further explanation after the mentioned sentence by the reviewer, between lines 405-411 of the main text, together with a remark that should the IFR decrease by a specific amount, the fraction of people infected would also increase. 

2. Table S1 in the Supporting Information (SI) shows the proposed method calculated point estimates for incubation period as 2.57 days, recovery period as 4.12 days, and transmission period as 1.7 days. These estimates appear to be lower than other resource indicates. For example, WHO’s COVID-19 Situation Report 73 indicates the incubation period for COVID-19 is on average 5-6 days. A study (vanKampen, June 2020) indicates the median duration of infectious virus shedding is 8 days post onset of symptoms. Please comment on how these epidemic parameters estimated from the proposed model compare to results from other studies and whether biases exist for these estimates.

 Response: We thank the reviewer for providing these additional references. We believe that these studies are well carried out and do not present any bias. We have introduced and commented in the last paragraph in page 8 (SI) the difference between our estimated parameters and those reported in these references. In principle, the difference stems from the fact that our calibration, which uses the Bayesian melding, accounts for all infected individuals (pre-symptomatic, asymptomatic, and symptomatic) regardless whether they are detected or not, and not only for those infected who are detected as in the references. Indeed, similar parameter values as those reported by the WHO and van Kampen et al (2020) have been used in our likelihood L_1 (θ) –see Eqs (6) and (8). However, after accounting for all infected individuals, and not just those individuals who are infected and detected, our posterior distribution gives that these parameter values are much lower.

3. I have following comments to the external validation results (i.e., Figure S6 and Table S5) for estimated IFR in the SI.

a. Figure S6 only shows the absolute difference between estimate and observed IFR in the right-hand side panel. It will be informative to also provide relative difference between estimated and observed IFR (i.e., relative difference=(estimated-observed)/observed).

 Response: Following the suggestion of the reviewer, we have included two additional panels in Figure S6. The absolute and the relative errors between the estimated fraction of people ever infected and the seroprevalence studies relative to the seroprevalence estimate. 

b. The labels in Figure S6 could be misleading, especially in the right-hand side panel, where -0.1 and 0.1 represents ±10% difference in fatality rate. Please change the label in x and y axis to percentage (e.g., -10%, +10%), which is consistent with Table S5 and rest of the document.

 Response: We thank the reviewer for this suggestion. We have re-labelled the axes accordingly.

c. For Table S5, please add 90%CI and/or 68%CI for the estimated IFR. This will illustrate how many of the observed IFRs actually fall in the credible intervals.

 Response: Thank you. We have included two columns with the 90% CI and the 68% CI. Also, we have included an additional column with the relative errors.

d. Some results in Table S5 show relatively large difference between estimated and observed IFR (e.g. Sao Luis, Brazil; Recife, Brazil; and Sao Paulo, Brazil) even with a large number of deaths. Please comment.

 Response: Thank you for your remark and allowing us to further develop on this point. The differences between the estimated and observed IFR are most probably driven by the particular serological survey design and the issues faced in conducting the study in the Brazilian context. Despite the individuals and households being randomly selected, the amount of surveys conducted were the same across all cities (between 200 and 250), irrespective of population size. Hence, extrapolating from a sample of 250 individuals to the city level in a place like São Paulo with 12.3 million inhabitants could be affecting the results by increasing the uncertainty levels. Nonetheless, we needed seroprevalence studies that included all age groups of the population and that were population level random studies (not from blood donors) in order to validate the model. This type of study is still very limited, as most seroprevalence surveys have either concentrated on specific subgroups of the population (e.g. health professionals) and specific age groups (often times excluding individuals younger than 14 or older than 65) or relied on blood donors (which provides a biased window on the overall infection in the population). Additionally, it is important to highlight that in a context of overwhelming uncertainty, seroprevalence studies are still the best resource we have to compare our estimates against, but they also have limitations, as we discussed in lines 70-84 in the Introduction of the main manuscript. We have added further detail on this study in the external validation part of the Supporting information (Section 3, page 11) to make it clearer to the reader the characteristics of this seroprevalence study. The reviewer can find the detailed changes marked in red.

---

## [Editor Report · Decision Letter 1]

11 Jan 2021

An indirect method to monitor the fraction of people ever infected with COVID-19: an application to the United States

PONE-D-20-30049R1

Dear Dr. di Lego,

We’re pleased to inform you that your manuscript has been judged scientifically suitable for publication and will be formally accepted for publication once it meets all outstanding technical requirements.

Kind regards,

Sanjai Kumar

Academic Editor

PLOS ONE
---

## [Editor Report · Acceptance letter]

19 Jan 2021

PONE-D-20-30049R1 

An indirect method to monitor the fraction of people ever infected with COVID-19: an application to the United States 

Dear Dr. di Lego:

I'm pleased to inform you that your manuscript has been deemed suitable for publication in PLOS ONE. Congratulations! Your manuscript is now with our production department. 

Kind regards, 

on behalf of

Dr. Sanjai Kumar 

Academic Editor

PLOS ONE